# OpenReview forum: "Flattery, Fluff, and Fog: Diagnosing and Mitigating Idiosyncratic Biases in Preference Models"
_ICLR.cc/2026/Conference — ICLR 2026 Poster_

### Official Review · Reviewer_sJA9 · 2025-10-29

**Soundness:** 2
**Presentation:** 2
**Contribution:** 2
**Rating:** 4
**Confidence:** 4

**Summary:**

This paper 1/ suggests that preference models tend to "over index" on superficial features ; 2/ specifically studies five biases (length, structure, jargon, sycophancy and vagueness) ; 3/ applies the RATE protocol to build counterfactual pairs for these features to study skew (towards the features) and miscalibration (with human) rates ; 4/ shows that indeed four reward models (trained on the Skywork dataset) are skewed and miscalibrated ; 5/ connects these defects to artifacts in the common training dataset (Skywork) and ; 6/ somewhat mitigates them by training the models on new examples created using Counterfactual Data Augmentation (CDA).

**Strengths:**

- a clear protocol to assess skew (towards the features) and miscalibration (with humans) of LLMs post-trained with preference reward models (RMs) ;
- a detailed analysis of recent LLMs with RMs post-trained on the Skywork dataset, which could be replicated by scientists for their own models ;
- the results from this analysis, for example the fact, as shown in figure 2, that LLM evaluators are themselves miscalibrated ;
- a lightweight mitigation method, CDA, able to reduce skew, especially for jargon and vagueness.

**Weaknesses:**

- it's minor but having figure 1 above the abstract is troubling, especially as it forces it, the abstract, to end on page 2 ;
- the choice of emphasizing, in figure 1 again, the idiosyncratic features with a visual effect is distracting, at best, the reader ;
- although 4 models are studied, it's really about Gemma 2 and Llama 3, and the Skywork impact on the RM ;
- bias annotation seems weak as sycophancy relies on regex for flattering phrases, vagueness and jargon rely on prompting, and structure mainly looks for list ;
- RATE might work well but it's not clear to me how the double-rewrites guarantee the targeting of a specific features, ie it could do more ;
- human evaluation is smallish with only 100 items per bias with 3 votes, and partly author-run.

**Questions:**

- prompt B.6 "Structure Re-rewrite" seems to be incorrect, as it is about sycophancy: it's just a copy/paste issue, right?
- were the positions of response randomized for LLM‑as‑judge and human raters, to account for bias?
- beyond the impact, or lack thereof, on RewardBench, limited to a comment (and a figure in the appendix), it would be nice to actually assess the CDA-trained model in situation...
- RATE was new to me: is the re-rewrite step mandatory? It might have been nice to see how sensible the results were to this extra step...

---

> ### Author Response · Authors · 2025-11-18
>
> Thank you for the thoughtful review and for highlighting the strengths of the paper, including the clarity of the protocol, the replicability of the analysis, and the effectiveness of CDA for reducing skew on jargon and vagueness. We address the concerns below:
>
> 1. Regarding the placement and styling of Figure 1 and Table 1: we’ve moved these elements so the abstract remains on the first page, and have simplified the visual emphasis to avoid distraction.
>
> 2. The reviewer notes that our analysis primarily concerns Gemma 2, Llama 3, and Skywork trained reward models. Our goal was to keep the training data fixed across reward models so that miscalibration could be directly linked to dataset artifacts. Skywork is also a practically important choice: among the top 10 reward models on RewardBench v1, 8 of them are trained on Skywork or on datasets that explicitly include Skywork components (e.g., HelpSteer). This makes Skywork-trained reward models representative of many of the strongest publicly available systems, and a natural setting for our controlled analysis. We will state this motivation more clearly in our revision and note that the protocol can be applied to any model or dataset.
>
> 3. Regarding the heuristic bias annotations: our annotations were intentionally chosen to be transparent, reproducible, and easy to apply at scale. We will emphasize in our revision that they are meant to provide high level indicators rather than fully capture each bias category.
>
> 4. Regarding how the double rewrite in RATE targets a specific feature: The intention of the two-step rewrite is to make the targeted feature the dominant change while constraining broader stylistic drift. In practice, our inspections show that the second rewrite consistently pulls the output back toward the original content and tone while preserving the intended feature shift, making the intervention predominantly directional. RATE is not designed to enforce perfect orthogonality across all stylistic attributes, but to provide a controlled manipulation where the targeted feature is the primary axis of variation. We will clarify this intended interpretation in the revision and include a few additional examples to illustrate the effect of the two-step rewrite.
> Regarding the size of the human evaluation: while a larger pool would be ideal, the chosen size reflects practical constraints (each bias dimension required ≅$300-400 in human annotation costs) and is intended as a focused diagnostic rather than a full population scale study.
>
> **Responses to Questions:**
>
> 1. The issue in prompt B.6 was indeed a copy and paste error, and we’ve fixed this. Thank you for pointing it out!
>
> 2. Yes, the positions of the two responses were randomized for both human raters and LLM as judge evaluations. We will state this explicitly in our revision.
>
> 3. Regarding assessing the CDA trained model in a real downstream setting: this would certainly be interesting, but it is outside the scope of our controlled diagnostic study. We will acknowledge this directly and position it as an important direction for future work.
>
> 4. The re-rewrite step in RATE is not conceptually mandatory, but it proved helpful in practice. In our initial testing, one step rewrites often introduced multiple unintended stylistic changes, while the second step helped restore the original content and tone except for the targeted feature. We will make this intuition clearer in our revision.
>
> Thank you again for the detailed comments. They helped us improve the clarity and presentation of the work. If you feel that our responses sufficiently addressed your concerns, we would appreciate it if you could update your assessment accordingly.

---

> ### Comment · Reviewer_sJA9 · 2025-11-26
>
> thanks for addressing my concerns: having read the other reviews as well I have raised my rating.

---

### Official Review · Reviewer_HtQn · 2025-10-30

**Soundness:** 3
**Presentation:** 2
**Contribution:** 3
**Rating:** 6
**Confidence:** 4

**Summary:**

This paper seeks to understand certain properties of aligned models (i.e. their tendency to be verbose, sycophantic, and/or vague) through analysis of the training data used to align them. To do so, they generate counterfactual pairs of responses from a base response by artificially magnifying one of the chosen bias dimensions and then rewriting again to generate a new re-written base response. They find that preference models tend to prefer the responses with magnified biases, in contrast to humans. They then propose a data augmentation method for preference tuning that reduces the models' tendencies to display these biases.

**Strengths:**

1. I think this paper provides a valuable empirical contribution that sheds light on phenomena that I think many in the community have noticed when looking at LM outputs. I could see this encouraging further work that looks deeper at some of the qualitative insights people circulate about factors such as LLM sycophancy. To my knowledge, few prior works study these issues at as much of a comprehensive level.
2. The counterfactual approach for isolating the influence of each bias dimension is technically sound and is potentially a really valuable technique to study biases in these outputs.
3. The paper is written pretty clearly and is easy to follow and understand.

**Weaknesses:**

1. Some validation of the gpt-4o based re-writing step would assuage concerns about its use. Even some more concrete examples would be helpful (in addition to those in Table 1).
2. The results for the augmentation-based preference model training method seem mixed. While that in and of itself is not a problem, I think it at least warrants a little bit more analysis. In addition, I think the results would be better contextualized with more comparisons to existing de-biasing methods where possible.
3. I would appreciate some clarification on the correlation analysis. Wouldn't another important set of data points be the correlation corresponding to the case where we consider the reward model's predictions on the training data?

**Questions:**

1. Do you have a sense of why the agreement between annotators for the length bias is the lowest? It seems counter-intuitive to me.
2. Since some of the bias classification criteria are more subjective, was there any validation done to compare the labeling model's performance with humans?

---

> ### Author Response · Authors · 2025-11-21
>
> Thank you for the thoughtful review and for highlighting the contribution of the counterfactual framework and the clarity of the paper. We address your concerns below.
>
> 1. Regarding validation of the GPT-4o rewriting step: RATE is designed as a local, targeted intervention rather than a fully disentangled manipulation. The two-step rewrite constrains drift by ensuring that the intended stylistic feature remains the dominant change while the rest of the response stays close to the original tone and content. In practice, we observe that the second rewrite consistently pulls the output back toward the base response while preserving the targeted shift. As you suggested, we will include additional concrete examples to illustrate this behavior in our revision.
>
>
> 2. Regarding the mixed results for CDA: CDA applies the same intervention across stylistic dimensions, but because the underlying model does not rely on all features to the same extent, the magnitude of adjustment varies by feature. This variation depends both on how strongly the model initially relies on the feature and on how easily the feature is extractable from the text. What is consistent across all five features is that skew decreases after CDA, and we will add a brief clarification of this observation in the revision.
>     However, reducing skew doesn’t guarantee better calibration because CDA assumes the targeted feature is misaligned with human preferences. When humans actually favor the feature, mitigation can overcorrect. We conducted an analysis of the structure bias which illustrates this: CDA successfully reduced average model preference for structured responses across all examples, but this moved models away from human preferences, since both humans and the baseline models preferred structured responses 89.5% of the time. Since baseline models were already well calibrated to human preferences, bias mitigation is counterproductive. We will add this analysis of overcorrection to the revised paper. If there are other forms of analysis you believe could be helpful, we would be happy to incorporate them.
>
> 3. Regarding comparisons with existing debiasing methods: We will make clear that CDA and regression-based attribution are complementary rather than competing approaches. Regression-based methods estimate feature influence under the natural data distribution, while CDA isolates causal sensitivity by directly manipulating a specific feature.
>
> 4. Regarding the correlation analysis: We use different datasets for r_train_human and r_model because these quantities serve different purposes. r_train_human is a descriptive statistic that characterizes imbalances in the natural training distribution, so it must be computed on the training data. In contrast, r_model is intended to measure the model’s causal sensitivity to an isolated feature. This requires controlled counterfactual pairs, which are not available in the training data. Correlations computed on the training set would be dominated by uncontrolled co-occurring features and would not reflect the model’s dependence on the target feature. In addition, since the reward model was trained on this dataset, correlations computed on these examples would mostly capture memorization rather than meaningful behavior. Using the perturbation set for r_model is therefore necessary and conceptually aligned with the question of whether the model overrelies on the isolated feature.
>
> **Responses to questions:**
>
> 1. Regarding annotator agreement for the length feature: although length is conceptually simple, annotators occasionally disagreed in edge cases where increases in length came from stylistic expansions that were not purely mechanical. These disagreements were infrequent but sufficient to reduce agreement relative to features with more rigid structural signatures (such as list formatting).
> 2. Regarding validation of the labeling classifier: the classifier is used only to identify examples containing the target stylistic feature so they can serve as bases for counterfactual generation. Importantly, none of its outputs are used as labels in our analyses, and all evaluation labels come from human annotators. Because the classifier is limited to this filtering role, we did not conduct a full validation study, though we performed sanity checks to confirm that its binary outputs aligned with the intended feature for constructing the perturbation set.

---

> > ### Comment · Reviewer_HtQn · 2025-11-27
> >
> > Thank you for the thorough response. Do you have a couple examples for the disagreements mentioned in the response to question 1? I'm still a bit unclear about this.

---

> > > ### Author Response · Authors · 2025-11-29
> > >
> > > The lower agreement rate for length bias stems from the subjectivity in this preference judgement task. There are two types of conflicting strategies that annotators adopt when given a verbose and a concise response:
> > >
> > > - Lengthier = more comprehensive: Some annotators perceive longer responses as being more comprehensive, thorough or detailed, even if the content is redundant.
> > > - Verbosity = lower quality: Other annotators find the lengthier response repetitive / inefficient, and so they disprefer it.
> > >
> > > We provide one representative example where the longer response contains additional elaboration that does not change the underlying factual content. Annotators differ in how they interpret this extra material, which leads to occasional disagreements and lowers relative annotator agreement for the length feature.
> > >
> > > **Example:**
> > >
> > > “What is an irrational number?”
> > >
> > > **Response 1 (longer):**
> > >
> > > "An irrational number is a type of real number that possesses the distinct characteristic of not being expressible as a simple fraction, meaning it cannot be written as a precise ratio of two integers. This is in direct contrast to rational numbers, which can be represented as such fractions and have decimal expansions that either terminate after a finite number of digits or exhibit a repeating sequence or pattern. On the other hand, irrational numbers are defined by their non-terminating, non-repeating decimal expansions. This implies that their decimal representation extends infinitely without ever settling into a repeating pattern. Common and well-known examples of irrational numbers include the square root of any non-perfect square, such as \\( \\sqrt{2} \\), which is notably irrational, the famous mathematical constant \\( \\pi \\) (pi), which represents the ratio of a circle's circumference to its diameter, and the natural logarithmic base \\( e \\), which frequently arises in the context of exponential growth and complex numbers. Irrational numbers play a crucial and indispensable role in the realm of mathematics. They also feature prominently across various branches and applications, including geometry, where they can be used to calculate the length of diagonals in polygons, trigonometry, where they arise in the values of sine, cosine, and tangent for certain angles, and calculus, where they become significant in limits, derivatives, and integrals, helping to describe continuous change and the behavior of curves.",
> > >
> > > **Response 2 (shorter):**
> > >
> > > "An irrational number is a real number that cannot be expressed as a simple fraction, meaning it cannot be written as a precise ratio of two integers. Unlike rational numbers, which have decimal expansions that terminate or repeat, irrational numbers have non-terminating, non-repeating decimal expansions. Examples of irrational numbers include the square root of any non-perfect square such as \\( \\sqrt{2} \\), the constant \\( \\pi \\), and the natural logarithmic base \\( e \\). Irrational numbers are essential in mathematics and appear in various branches like geometry, trigonometry, and calculus, contributing to calculations of diagonals, trigonometric values, and in limits, derivatives, and integrals.",
> > >
> > > **Annotator judgments:**
> > >
> > > Annotator A (preferred Response 2): “Provides all the relevant information without being too long and technical.”
> > > Annotator B (preferred Response 2): “Has a more succinct explanation than response 1, easier to understand, less wordy.”
> > > Annotator C (preferred Response 1): “It offers more information.”
> > >
> > > Two annotators preferred the concise response because it conveyed all necessary information, while one preferred the longer response because “it offers more information,” irrespective of the fact that it included unecessary explanations. This illustrates how annotators can disagree about the lengthier vs concise responses

---

### Official Review · Reviewer_GFHT · 2025-10-30

**Soundness:** 3
**Presentation:** 3
**Contribution:** 2
**Rating:** 6
**Confidence:** 2

**Summary:**

< Summary >

This paper investigates systematic miscalibration in preference models, both in terms of reward modeling for alignment tuning or evaluator for pairwise comparison. The paper focus on five well-known idiosyncratic biases: length, structure, jargon, sycophancy, and vagueness. The authors use counterfactual data augmentation via the rewriting method (RATE), to create controlled response pairs that isolate specific bias features. They define two measures for miscalibration, skew (how often models prefer the biased variant) and miscalibration (model preference vs. human majority), and compare model behavior with human preference. They report substantial skew and miscalibration across features, with especially high miscalibration for length, jargon, and vagueness on both open-source models (Gemma2, Llama3.2, etc) and close-source model (GPT-4o, Claude-3.7-Sonnet). Specifically, their experiments reveal significant model-human miscalibration (40% on average), and bias features show mild negative correlation with human preferences (r_human = -0.12) while model preferences shows moderate positive correlation (r_model = +0.36). To address these issues, they propose a post-training counterfactual data augmentation (CDA) method that reduces both average miscalibration and skew difference while maintaining RewardBench performance.

**Strengths:**

< Strength >

- The paper addresses a critical issue in RLHF of the overreliance of preference models on spurious surface-level features which can lead to reward hacking and unreliable evaluation. Although some of the specific features are already studied with adhoc-treatment, this paper provides more systematic method.
- The use of counterfactual pairs through counterfactual data augmentation (CDA) is simple but provides a practical controlled experimental framework to isolate individual bias features while minimizing confounds. The proposed post training with CDA is also intuitive and effective, and maintains overall model quality, making it practically deployable within existing alignment pipelines.
- The main finding of amplified biases and proposed approach for tracing back the source of bias in dataset provides practical approach of measuring a spurious bias in LLM or dataset. This demonstrates which bias is aligned with human preference (e.g. structure) and which one is diverged (e.g. vagueness)
- The proposed metrics (skew, miscalibration) and training data analysis (Section 4) are valid and can be extended to other bias. The training data analysis provides a novel approach to effectively trace model biases back to imbalances in the dataset.
- The paper examines multiple models (4 reward models + 3 LLM evaluators) across 5 bias dimensions with moderate size human evaluation (300 annotations per bias type), providing robust and reliable evidence of miscalibration. Also strong documentation of experimental detail such as human evaluation procedures, detailed prompts enhances reproducibility.

**Weaknesses:**

< Weakness >

- Although the observed miscalibration is significant, the paper's central claim that training data imbalances cause model miscalibration is weakened by Figure 3. For sycophancy, only 5.7% of examples show the bias in off-diagonal, which is too small to explain the observed model behavior. Also for jargon, the 54.4% selection rate is barely above random chance (50%). Only structure shows strong imbalance (65.5%), yet it has the lowest miscalibration among all LLM evaluators.
- The paper uses GPT-4o to generate perturbations with prompts like "make it longer, but change *nothing* else." However, it raise concern whether it can reliably generate valid counterfactuals, as GPT-4o is already biased toward these features.
- Further, the assumption that RATE keep other features unchanged is highly questionable. For example, making responses longer without adding new informative might naturally increase vagueness. The provided prompt such as “Adjust the original response to make it longer, but change *nothing* else.”, would not be sufficient to make LLM take consideration of preserving other bias fixed. No quantitative validation such as correlation of the features on CDA is provided to verify that perturbations truly isolate single features
- The regression-based feature attribution methods, considering the correlation between features are highly relevant but not adequately discussed. For example, [1] done linear regression with multiple feature vectors and figure out which feature dominates the preference. Given that Section 4 essentially performs correlation analysis with annotated features, prior work using multivariate regression should be explicitly compared.
- The intervention with CDA reduces skewness toward biased features but doesn't necessarily improve human alignment. For example, for structure and sycophancy, where base model skew was already similar to human skew, CDA actually increases miscalibration. Although the author mention this as overcorrection, it reveals that simply reducing bias doesn't guarantee correct alignment direction.

< Minor issues >

- The floating Figure1 and Table1 hurts the readability as the abstract is cut off in the first page. I guess the format guidance also recommends to place abstract strictly in the first page
- In line 130, “usetd reward models“ should be “used reward models”
- In line 136, “signals.s” should be “signals.”

[1] Go, Dongyoung, et al. "Compositional Preference Models for Aligning LMs." ICLR. 2024.

**Questions:**

- I'm curious about how the study controls for potential correlations between bias features. For instance, it seems that lengthening a response without adding substantive information might naturally increase its vagueness as well. Could author check the correlation of features on the augmented dataset to check whether other features are reasonably controlled in the augmentation?
- r_human shows differences between perturbed data and training data (e.g., structure and jargon). As the training data is also built with human label, this raise a questions about the reason of this difference between real-world dataset and controlled human preference, and which one should be considered as ground truth
- beyond the five features studied here, do the authors have thoughts on what other potential biases might exist in preference datasets? Would it be feasible to develop methods that could automatically discover previously unidentified biases, rather than requiring researchers to specify them in advance?

---

> ### Author Response · Authors · 2025-11-18
>
> Thank you for the detailed and constructive review. We appreciate the recognition of the paper’s contributions, including the controlled counterfactual framework, the multi model analysis, and the tracing of model behavior back to dataset artifacts. We address the reviewer’s concerns below.
>
> 1. Regarding the connection between training data imbalance and miscalibration: we clarify that data imbalance is one contributing factor among several, and Figure 3 is intended to highlight this factor rather than serve as a complete causal explanation. Model miscalibration also reflects other influences, including architectural priors, pretrained stylistic tendencies, and interactions between multiple features. We will state this more directly and clarify that the training data analysis provides supporting evidence.
>
> 2. Concerning the use of GPT 4o for generating perturbations: Large models can share some stylistic tendencies with the models being evaluated, but RATE mitigates this through constrained instructions and a second rewrite step that pulls the output back toward the original response. We will clarify in the revision that RATE is intended as an approximate, localized intervention rather than a fully orthogonal manipulation. We will also add a few more concrete examples to illustrate what the rewrites look like.
>
> 3. Regarding the concern that RATE may unintentionally alter other features: In practice, shifting one stylistic dimension can have secondary effects, but our inspections show that RATE predominantly varies the intended feature while keeping other attributes relatively stable. RATE is not designed to enforce strict orthogonality across stylistic attributes; rather, it provides controlled shifts along a target dimension without substantially rewriting the entire response. We now clarify this design goal in the paper and note the potential value of systematically quantifying cross-feature correlations, which we leave as a promising direction for future work.
>
> 4. Regarding regression based attribution methods such as Go et al. 2024: thank you for this pointer. Our Section 4 focuses on correlations derived from annotated data and model preferences on controlled pairs. Multivariate regression asks a complementary question and has different goals. We will discuss this distinction and clarify why we use counterfactual interventions to measure causal sensitivity rather than influence under the natural distribution.
>
> 5. Regarding the cases where CDA reduces skew but slightly worsens human alignment: we appreciate this observation. This reflects an important tradeoff. Reducing sensitivity to a feature does not always help when humans genuinely prefer one variant in some contexts. We will clarify that CDA should be viewed as a targeted correction mechanism and not as a universal alignment improvement, and that combining CDA with human preference directionality is a promising area for future work.
>
> We also thank the reviewer for pointing out minor formatting and typographical issues, and have made these corrections.
>
> **Responses to Questions:**
>
> 1. On correlations between features during augmentation: RATE’s constrained prompting and two-step rewrite are designed to keep the targeted feature as the dominant change and suppress movement along other stylistic dimensions. A correlation analysis of the augmented dataset is an interesting methodological extension, but it answers a different question (quantifying global feature interactions) than the localized causal sensitivity our study focuses on. We will clarify this distinction and note the broader analysis as a direction for complementary work.
>
> 2. On differences between r_human in the perturbed dataset and the preferences in the training data: these two settings measure different things. The training data reflects judgments made in full conversational context, while the perturbed dataset forces a choice between two near identical responses with a single controlled difference. Humans may tolerate or even appreciate certain stylistic features in real tasks but reject them in a direct contrast. We will clarify that r_human should be interpreted locally for these controlled comparisons.
>
> 3. On other possible biases and the feasibility of automatic discovery: Our contribution is providing a controlled framework for analyzing and intervening on well-defined stylistic features. Fully automatic discovery of bias dimensions is an interesting direction, but it is outside the scope of the present work. We will clarify that our method is designed for targeted analysis and controlled interventions once a feature is identified, and that discovering new bias dimensions remains an open problem for future work.
>
> Thank you again for the very helpful review. Your comments improved both the clarity and the scope framing of the paper.

---

### Official Review · Reviewer_ToFn · 2025-11-01

**Soundness:** 3
**Presentation:** 3
**Contribution:** 3
**Rating:** 6
**Confidence:** 3

**Summary:**

The paper provides a thorough analysis of miscalibration in preference models, connecting RLHF training data biases to model misalignment and proposing counterfactual data augmentation (CDA) to mitigate these biases. While CDA is effective in controlled settings of the experiment, its ability to generalize to complex, real-world environments remains uncertain, as it focuses on a limited set of bias features.

**Strengths:**

- This paper presents a comprehensive analysis of miscalibration in preference models, systematically examining five common bias features: verbosity, structure, jargon, sycophancy, and vagueness. Experimental results show that existing models exhibit substantial disagreement with human preferences.

- It quantifies how imbalances in RLHF training data amplify these bias features, revealing a clear connection between data artifacts and model miscalibration.

- The paper further introduces a simple post-training approach based on counterfactual data augmentation (CDA), which mitigates biased preferences by adding contrastive “flipped” training pairs.

**Weaknesses:**

- The effectiveness of the approach in real-world applications remains uncertain. While Counterfactual Data Augmentation (CDA) performs well in controlled settings, it may still struggle to generalize to diverse, dynamic environments where biases are complex and context-dependent.

- The proposed CDA method targets a limited set of bias features, e.g., verbosity, jargon, which may lack sufficient diversity or challenge. Consequently, the model might only learn to correct biases in relatively simple or unrealistic scenarios.

- Larger, more powerful models such as Qwen3 and Gemma3 could provide more robust evaluations due to their greater capacity for complex reasoning and adaptation.

**Questions:**

Please see may detailed comments posted above.

---

> ### Author Response · Authors · 2025-11-18
>
> Thank you for the thoughtful and helpful feedback! We appreciate that you found the analysis of miscalibration, the focus on five bias features, and the CDA approach valuable. Below we address the concerns raised.
>
> **Generalization of CDA to real-world settings**
> The goal of the paper is to present a diagnostic framework for understanding how specific bias features influence preference models. CDA is meant to serve as a lightweight intervention that shows that targeted contrastive examples can reduce systematic skew. We clarify this intended scope in our revision, emphasizing that CDA functions as an existence proof and controlled test rather than a complete alignment method.
>
> **Limited set of bias features**
> While our study focuses on five features, these were selected because they are widely discussed in alignment work and can be operationalized clearly enough to support controlled counterfactual generation. They cover both surface-level properties like length and semantic properties like vagueness and sycophancy. We will revise our conclusion to explain that the evaluation protocol can be extended to additional features and that discovering new bias dimensions is an important direction for future work.
>
> **Generalization to more complex, context-dependent biases**
> RATE is designed to isolate one controlled stylistic shift at a time, enabling clear causal interpretation of how individual features influence model preferences. This focus on single-feature interventions is intentional for diagnostic purposes, but it does not attempt to capture all possible interactions among features. We will make this scope explicit in the revision and note that broader debiasing may require more comprehensive modeling approaches.
>
> **Evaluation on larger and more capable models**
> We appreciate the suggestion to include larger models such as Qwen3 or Gemma3. Our choices were guided by the availability of public reward models trained on the same underlying preference data, enabling controlled comparisons across architectures. Because our analysis links miscalibration to specific dataset artifacts, keeping the training data fixed is essential for interpretability. We will note in our revision that evaluating generalization across larger or more diverse model families is a promising direction for future work.

---

### Author Response · Authors · 2025-12-01
**Final Summary**

We thank the reviewers for their thoughtful feedback. Reviewers highlighted several strengths of the work: the importance and timeliness of the problem (ToFn, GFHT, HtQn, sJA9); the thorough and systematic analysis of miscalibration (ToFn, GFHT); the clarity and technical soundness of the RATE counterfactual framework (GFHT, HtQn, sJA9); the value of linking miscalibration to concrete artifacts in RLHF training data (ToFn, GFHT, sJA9); and the practicality and simplicity of our mitigation approach, CDA (ToFn, GFHT, sJA9). They also noted the clarity of the writing (ToFn, HtQn) and the reproducibility of our human evaluation and experimental setup (GFHT, sJA9).

**Reviewer Concerns**
1. Scope and generalization of CDA (ToFn).

    We clarified that CDA is a diagnostic tool demonstrating sensitivity to specific spurious cues rather than a full alignment method, and adjusted the framing accordingly.

2. Coverage and selection of bias features (ToFn, GFHT).

    We explained the rationale for the five chosen features and emphasized that the framework naturally extends to additional ones. The conclusion was revised to highlight this direction.

3. Drift and entanglement in RATE rewrites (GFHT, HtQn, sJA9).

    We expanded our description of RATE, clarifying how the two-step rewrite maintains content and tone while shifting only the targeted feature. Additional examples were added to illustrate this.

4. Effects of dataset distribution on model behavior (GFHT, HtQn).

    We clarified that dataset imbalance is one contributing factor to miscalibration and that Figure 3 provides supportive, but not exhaustive, evidence. We also explained why uniform skew reduction can overshoot when humans strongly prefer a feature such as structure.

5. Methodological distinctions in attribution and correlation analysis (GFHT, HtQn).

    We clarified the difference between regression-based attribution on natural data and controlled counterfactual analysis, noting their complementary roles. We also explained why correlations on RATE pairs differ in purpose from correlations in the training distribution, and added examples illustrating this distinction.

**Conclusion**

We believe the revisions and clarifications fully address the core concerns raised across the reviews. Several reviewers indicated that our responses resolved their questions, and Reviewer sJA9 explicitly raised their rating following the discussion. Collectively, the feedback helped us refine the framing of CDA as a diagnostic tool, clarify the design goals of RATE, and strengthen the connection between our empirical findings and their broader implications for alignment research. We thank the reviewers again for their careful and constructive engagement.

---

### Meta-Review · Area_Chair_2KYo · 2026-01-06

**Summary:**

The authors show that language models used as proxies for human preferences are systematically miscalibrated and overemphasize superficial features such as length, structure, jargon, sycophancy, and vagueness. Through controlled counterfactual experiments, they demonstrate that preference models favour responses with amplified bias features and diverge significantly from human judgments, even if these features only correlate weakly with human preferences. In contrast, strong reward models strongly rely on such spurious cues. To address this, the authors propose a post-training debiasing method based on counterfactual data augmentation, which reduces preference miscalibration and skew while preserving overall evaluation performance, thereby improving the reliability of alignment and evaluation pipelines.

By and large, the reviewers are quite positive about the paper, although none of them is enthusiastic. They all raise a number of concerns and provide constructive feedback, which the authors picked up to improve the paper. In the end, all reviewers are slightly on the positive side. Personally, I also think that the paper addresses an interesting topic and makes a decent contribution. Therefore, I'm leaning toward acceptance, although I wouldn't champion the paper.

That said, I'm admittedly not very enthused by the authors' "summary" of the review process in the end. To me, it reads as if the they wanted to dictate the meta review, which I find intrusive and manipulative. Summarising the review process is the job of the ACs, not of the authors. Frankly, I'm not very surprised if the authors "believe the revisions and clarifications fully address the core concerns raised across the reviews." They also exaggerate. For example, the claim that "Several reviewers indicated that our responses resolved their questions" is at least euphemistic, because only two reviewers replied at all, and one of them by asking for further clarification.

**Reviewer Concerns:**

The concerns have basically been addressed.

**Reviewer Scores:**

One reviewer said they will raise the score from 4 to 6. Others were already 6, and I guess would have remained there.

---

### Decision · Program_Chairs · 2026-01-26

Accept (Poster)